# Novel Multilayer SAW Temperature Sensor for Ultra-High Temperature Environments

**DOI:** 10.3390/mi12060643

**Published:** 2021-05-31

**Authors:** Xuhang Zhou, Qiulin Tan, Xiaorui Liang, Baimao Lin, Tao Guo, Yu Gan

**Affiliations:** Science and Technology on Electronic Test and Measurement Laboratory, North University of China, Taiyuan 030051, China; 18434361542@163.com (X.Z.); B1906058@st.nuc.edu.cn (X.L.); linbaimao163@163.com (B.L.); guotao6@nuc.edu.cn (T.G.); ganyu9608@163.com (Y.G.)

**Keywords:** high-temperature electrode, SAW sensor, AlN films, langasite

## Abstract

Performing high-temperature measurements on the rotating parts of aero-engine systems requires wireless passive sensors. Surface acoustic wave (SAW) sensors can measure high temperatures wirelessly, making them ideal for extreme situations where wired sensors are not applicable. This study reports a new SAW temperature sensor based on a langasite (LGS) substrate that can perform measurements in environments with temperatures as high as 1300 °C. The Pt electrode and LGS substrate were protected by an AlN passivation layer deposited via a pulsed laser, thereby improving the crystallization quality of the Pt film, with the function and stability of the SAW device guaranteed at 1100 °C. The linear relationship between the resonant frequency and temperature is verified by various high-temperature radio-frequency (RF) tests. Changes in sample microstructure before and after high-temperature exposure are analyzed using scanning electron microscopy (SEM) and X-ray diffraction (XRD). The analysis confirms that the proposed AlN/Pt/Cr thin-film electrode has great application potential in high-temperature SAW sensors.

## 1. Introduction

As science and technology advance, the requirement for functional materials suited to conducting research in ultra-high temperature environments, such as space exploration [1], aero-engine combustion chambers [2,3,4,5], geothermal research [6], and nuclear reactors [7], has intensified. Testing measurement systems in high-temperature environments is challenging, with accurate temperature measurements critical to achieve improvements [8]. As common passive wireless devices, surface acoustic wave (SAW) devices have attracted widespread attention because of their high operating frequency, digital compatibility, and high reliability [9,10,11]. Traditional SAW devices tend to show performance decline or even complete failure in high-temperature environments [12]. Notably, langasite (LGS) piezoelectric single-crystal materials offer unique advantages over traditional piezoelectric materials under such conditions. Compared with LiNbO_3_ and LiTaO_3_, LGS has a smaller temperature coefficient and greater stability at high temperatures. Additionally, compared with quartz crystals, LGS has a larger electromechanical coupling coefficient. The melting point of LGS is 1470 °C and it maintains a stable phase from room temperature up to its melting point, indicating that monocrystalline LGS piezoelectric materials can maintain stable piezoelectric performance at extremely high temperatures. Therefore, the conductive stability of electrodes at high temperatures has become a key factor for determining the suitability of SAW devices, and many researchers have studied the relationship between the conductive performance of metal electrodes at high temperatures and their microstructure. For example, Thompson [13] found that metal films agglomerate at high temperatures, resulting in a loss of electrical conductivity. In addition, Sakharov et al. [14] and Aubert et al. [15,16] prepared Ir-based composite electrodes and demonstrated their stable operation above 800 °C. Alternatively, Moulzolf et al. [17] prepared Pt-ZrO_2_ and Pt-HfO_2_ composite electrodes that operated stably for 4 h at 1000 °C. In addition, Aubert et al. [18,19,20,21] used Ta as the adhesive layer in the electrode to prepare a SAW device capable of stable operation at 1000 °C for at least 30 min.

In this paper, an AlN passivation layer is deposited on the SAW device to protect the piezoelectric substrate and the interdigital transducer (IDT) electrode, so as to improve the working temperature and working time of the interdigital electrode. This article introduces the design and manufacturing process of this high-temperature SAW sensor, as well as the preparation of the protective film, and the high-temperature experiment performed on the SAW device. A SAW sensor with an interdigital width of 4 μm can measure temperatures up to 1300 °C, while having good repeatability and durability at 1100 °C, which is a great improvement.

## 2. Materials and Methods

### 2.1. Sensor Fabrication

In this study, the manufacturing of SAW sensors is divided into two parts: the preparation of interdigital transducers (IDTs) of SAW sensors and the deposition of the protective layer. Typically, SAW resonators have a single-port configuration consisting of an IDT and two reflector sets. Each IDT contains 40 equidistant fingerlike electrodes, with 80 pairs of reflective grids placed on each side of the IDT. The IDTs and reflective fingerlike gratings have a width of 4 μm. An LGS substrate with an Euler angle (0°, 138.5°, 15°) was selected, and the IDTs were constructed from Pt, which has a melting point of 1770 °C. Metal Cr serves as the adhesion layer between Pt and LGS, which increases the adhesion between LGS and Pt and prevents the IDTs from falling off under high temperature. The preparation process of the sensor is shown in Figure 1. The SAW sensor manufacturing process includes substrate cleaning, homogenization, photolithography, and development. The SAW resonator is manufactured using ultraviolet photoengraving (EVG610, EV Group). Additionally, O_2_ plasma was treated with 60 W radio-frequency (RF) power for 6 min (IoN Wave 10, PVA TePla America, Inc., Corona, CA, USA) to increase metal adhesion before deposition. A Cr/Pt (20/200 nm) film was prepared as a fork-finger electrode for the SAW sensor using a magnetron sputtering system.

Next, AlN thin films were deposited on the surfaces of the IDTs and reflective grids to protect the SAW devices. Large electrode pads were exposed to high-temperature testing to connect the Pt wires. A 100 nm-thick AlN layer was deposited on the IDTs and the LGS substrates by pulsed laser deposition (PLD; PLD-450B, Shenyang Scientific Instrument). The laser has a constant voltage output of 22 kV, a frequency of 3 Hz, a deposition chamber pressure of 3.5 × 10^−4^ Pa, and a deposition temperature of 150 °C. The purity of the AlN target was 99.999%.

Before performing measurements, the sample was heated to 700 °C in a muffle furnace to improve its thermal stability. Thermal annealing not only improves the electrical properties of the Pt/Cr electrode at high temperatures, but also eliminates the voids in the AlN coating, thus improving the thermal stability of the prepared LGS SAW sensor. Figure 2 shows a schematic diagram of the SAW sensor, a photograph of the prepared sensor, and a schematic diagram of the sensor structure.

### 2.2. High-Temperature Measurements

Figure 3 shows the test scheme for the temperature sensor. The temperature dependence of the resonant frequency of the SAW temperature sensor was measured using an RF network analyzer (E5061B, Agilent) and a muffle furnace. Because the maximum test temperature of the SAW sensor is 1300 °C, it could not be attached to a printed circuit board (PCB). Therefore, the Pt wire was fixed on the fork finger electrode using Pt slurry, while the other end of the Pt wire was connected to the PCB, which, in turn, was connected to the network analyzer via a coaxial cable. When the frequency sweep signal sent by the network analyzer is consistent with the resonant frequency of the sensor, it resonates, and the sensor signals are transmitted back to the network analyzer through the connected Pt wire. The operating frequency of the sensor was determined by extracting the curve from the network analyzer. The heating rate of the muffle furnace was set at 10 °C/min, with measurements recorded from 100–1300 °C in 100 °C intervals through the K-type thermocouple to obtain more accurate real-time temperature values and the real-time sensor performance.

## 3. Results and Discussion

Muffle furnace temperature tests were conducted from 25–1300 °C using the S11 curves recorded for different temperature points by the network analyzer. As the experimental temperature continues to rise, the resonant frequency of the sensor continues to decrease, and the sensor can be tested to 1300 °C. When the temperature increases from 25 °C to 1300 °C, the resonant frequency of the SAW sensor decreases from 158.53 MHz to 155.59 MHz, and the resonant frequency changes 2.94 MHz. In the test temperature range (25–1300 °C), the average change of the sensor’s resonance frequency is 2.31 kHz/°C.

In addition, the resonance frequency changes nonlinearly as the temperature increases. As shown in Figure 4a, the sensitivity of the sensor gradually increases as the temperature increases. The relationship between sensor frequency and temperature is defined by fitting the experimental response curve. Because LGS has a non-zero frequency temperature coefficient (TCF), the TCF of LGS was close to zero at room temperature and became negative with increasing temperature; therefore, the resonant frequency of the sensor increased and then decreased, including an increase between room temperature and 100 °C. In the low temperature range (100–400 °C), a sensitivity of 0.72 kHz/℃ was observed, increasing first to 2.01 kHz/°C for the mid temperature range (400–900 °C), then to 4.38 kHz/°C for the high temperature range (900–1300 °C), as shown in Figure 4b. To reflect the change in sensitivity, we fitted different curves to the sensor response for different temperature ranges and created a piecewise function to improve the temperature estimation accuracy. The relationship between the resonant frequency and temperature in different temperature ranges is described by Equation (1), where *T* represents temperature and *FL*, *FM*, and *FH* are sensor frequencies in the low, mid, and high temperature ranges, respectively:*FL* = −0.000723*T* + 158.67098 (100–400 °C),(1)
*FM* = −0.00201*T* + 159.14844 (400–900 °C),(2)
*FH* = −0.00438*T* + 161.32726 (900–1300 °C).(3)

As the temperature increases, the change of the surface temperature of the LGS substrate will cause the elastic constant, piezoelectric constant, and dielectric constant of the substrate to change, thereby changing the speed of the surface acoustic wave propagating on it. The mechanism of the sensor is to influence the change of resonance frequency through the change of LGS surface acoustic wave velocity. In this way, any change in dielectric constant, piezoelectric constant, and elastic constant will affect the change in surface acoustic wave velocity. There are four different cationic lattice points in the LGS crystal structure. As the temperature increases, ions absorb heat energy, thereby increasing their random motion. With temperature changes, this irregular ion movement increases, and the piezoelectric constant and elastic constant change nonlinearly. The dielectric constant of LGS therefore increases with increasing temperature. At different temperatures, the degree of heat-induced molecular motion is different. Higher temperatures increase the thermal motion of molecules and increase the rate of change of dielectric constant. These may be the reasons for different temperature ranges and different sensor sensitivity.

However, the number of times the sensor can be tested at 1300 °C is limited. When the sensor was tested at 1300 °C for the fifth time, the resonant frequency of the sensor changed abruptly during the cooling process, indicating that the cross-finger electrode of the sensor had broken and raising the possibility of sensor damage. Since the working temperature of most turbine metal alloys does not exceed 950 °C, high temperature sensing in the range of 800 °C to 1100 °C is of special significance for the research and development of aeroengines. The sensor was tested three times at 1100 °C to analyze the stability of the resonant frequency response to temperature changes, as shown in Figure 5a. The repeatability of the sensor response is demonstrated by the consistency of the frequency curves recorded for the three measurement sets. The SAW sensor has been tested at 1100 °C for 1 h, and the performance of the SAW sensor remains stable, which shows that the SAW sensor has quite good durability.

Figure 6 shows the X-ray diffraction (XRD) results for the AIN/Pt/Cr/LGS samples before and after the high-temperature measurements. Before the measurements were performed, the XRD curve showed no distinct peaks. However, after the high-temperature measurements, a clear peak corresponding to Pt (111) appeared. There are three possible reasons for this: the Pt film did not crystallize, the crystallization quality was poor before the high-temperature measurements, or the platinum film crystallized after the high-temperature measurements. The results indicate that the Pt undergoes significant recrystallization when exposed to high temperatures, thus forming completely isolated grains and resulting in a significant increase in resistance, which affects the resonant frequency of the sensor.

After treatment at high temperature, the adhesion of the AlN/Pt/Cr film to the substrate was very good. After the first high-temperature test at 1300 °C, the frequency response did not deteriorate. This result is supported by scanning electron microscopy (SEM). As shown in Figure 7a, before the high temperature measurement, the surface of the sensor is smooth and there are no obvious particles. However, after a high temperature test at 1300 °C, it can be clearly found from Figure 7b,c that there are many small crystal grains on the inter-finger electrode. These small crystal grains are considered to be AlN particles. With stable conductivity, the sensor can continue to work. After the fifth high temperature test at 1300 °C, the surface morphology of the sensor was observed. As shown in Figure 7e,f, some large discontinuous crystal grains appeared on the surface of the sensor. The large crystal grains that appeared at this time originated from the Pt film. Due to the agglomeration and recrystallization of Pt, the interdigital electrode is broken, causing the electrode to lose conductivity and causing damage to the sensor. After testing at 1100 °C, the surface morphology of the sensor is shown in Figure 7d. The surface of the interdigital electrode is relatively smooth, with some small particles and no obvious large particles, which indicates that the sensor is in good condition and can still work continuously.

Table 1 presents a comparison of the sensor developed in this study with previously reported sensors [14,17,20]. The sensor proposed herein has the following advantages:The temperature range is large enough to monitor a relatively large temperature range.The sensor has a long working time at high temperature.Fitting is performed in three temperature stages, which improves the result of sensor estimation.

## 4. Conclusions

In this study, an AlN/Pt/Cr SAW temperature sensor was fabricated using micro-electromechanical systems (MEMS) technology. The sensor is capable of measuring a maximum temperature of 1300 °C, has good repeatability at 1100 °C and a high temperature endurance of not less than 1 h, and is approved for passivating protection of AlN thin-film covered LGS substrates and IDTs of SAW devices. The resonant frequency of the sensor drifts by 2.94 MHz over the observed temperature range. To improve the precision of the sensor estimation results, the temperature dependency of the sensor response from 25–1300 °C was divided into three parts, each with a different linear relationship between the temperature and the resonant frequency. As research progresses, remote wireless passive high-temperature testing will be realized by attaching the SAW sensor to a ceramic plate and bonding the helical antenna with Pt paste. The real-time operating temperature of turbine blades in high temperature and harsh environments can be directly measured by LGS-based wireless passive sensors. By continuously optimizing the size of the antenna, smaller size wireless transmission can be achieved. At the same time, for higher temperatures, new piezoelectric material SAW sensors need to be considered. Our method can be a good reference point to construct new devices.

## Figures and Tables

**Figure 1 micromachines-12-00643-f001:**
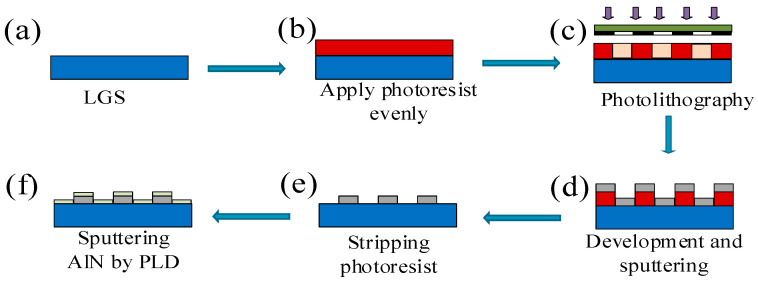
Schematic of the fabrication process of the sensor: (**a**) clean langasite (LGS) substrate; (**b**) apply photoresist evenly on the LGS surface; (**c**) lithography through reticle; (**d**) deposition of metal Pt/Cr by magnetron sputtering; (**e**) removal of photoresist; (**f**) AlN film deposition by PLD.

**Figure 2 micromachines-12-00643-f002:**
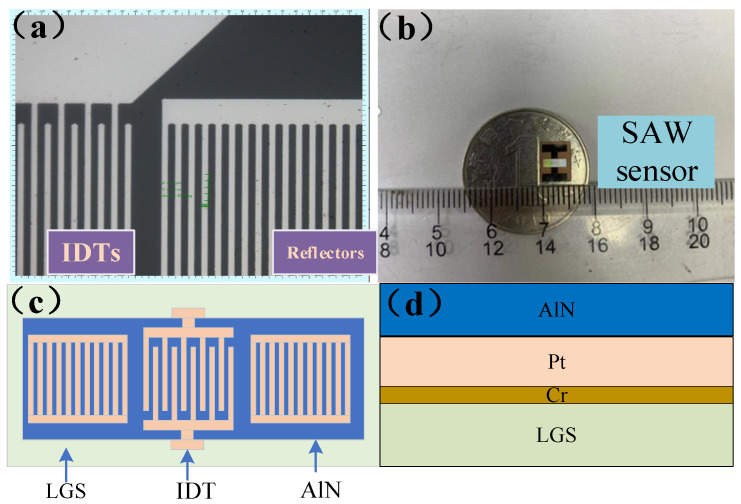
Structure design of surface acoustic wave (SAW) device: (**a**) the interdigital transducers (IDTs) structure; (**b**) SAW sensor; (**c**) diagram of sensor structure; (**d**) side view of the sensor structure.

**Figure 3 micromachines-12-00643-f003:**
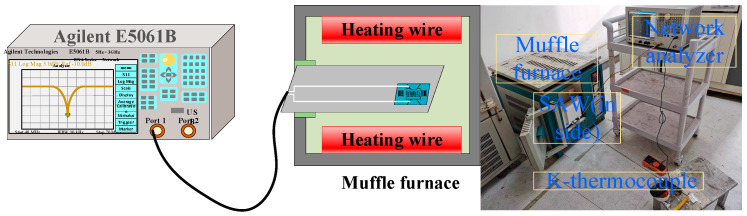
High temperature test system, real-time access to the S11 performance of SAW devices in temperature experiments.

**Figure 4 micromachines-12-00643-f004:**
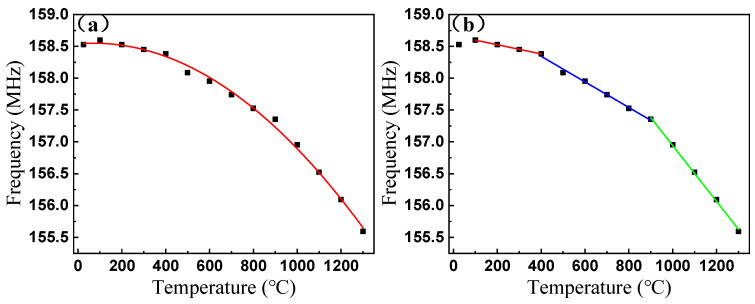
The frequency responsiveness of the SAW sensor during experiment: (**a**) the frequency responsiveness of the SAW sensor during 1300 °C experiment; (**b**) fitting curve for three temperature ranges.

**Figure 5 micromachines-12-00643-f005:**
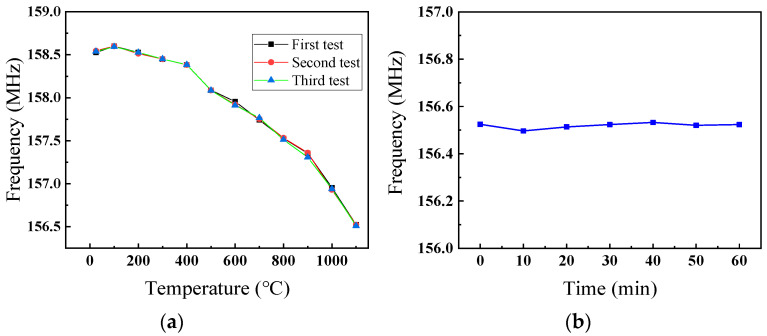
(**a**) The three repeatability experiments of extraction points and curves were between frequency and temperature; (**b**) frequency change graph of heat preservation sensor at 1100 °C.

**Figure 6 micromachines-12-00643-f006:**
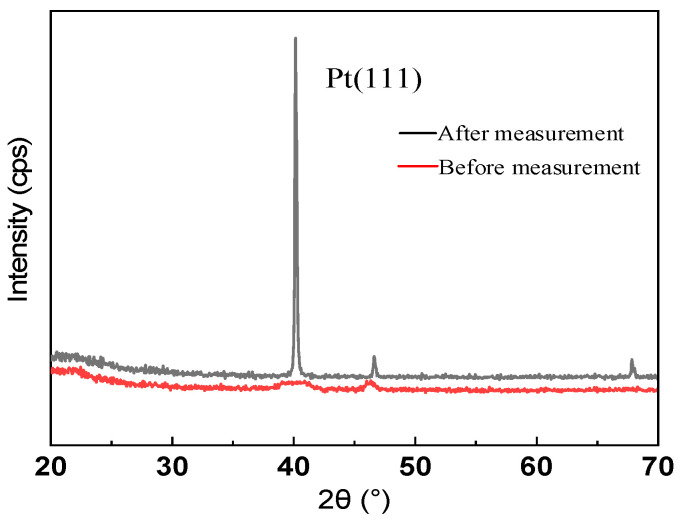
θ–2θ scans of AlN/Pt/Cr/LGS samples before and after high temperature measurement.

**Figure 7 micromachines-12-00643-f007:**
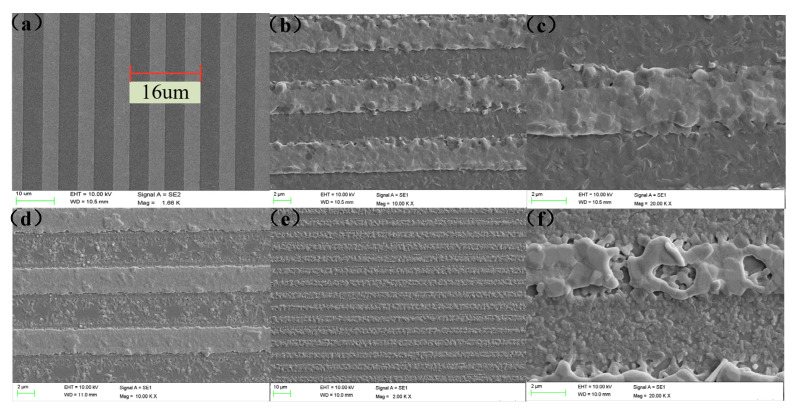
Surface topography of sensor samples: (**a**) before high temperature measurement; (**b**) grazing scanning electron microscopy (SEM) image of a AlN/Pt/Cr IDT after the first 1300 °C high temperature test; (**c**) partial enlarged view of IDTs; (**d**) grazing SEM image of AlN/Pt/Cr IDT after annealing at 1100 °C high temperature test; (**e**) grazing SEM image of AlN/Pt/Cr IDT after repeated annealing at 1300 °C high temperature test; and (**f**) partial enlarged view of IDTs.

**Table 1 micromachines-12-00643-t001:** Comparison between the sensors we studied and previously reported sensors.

Electrode Materials	Range	References
Pt/Ta	1000 °C, 30 min	[20]
Ir	800 °C	[14]
Pt-ZrO_2_ and Pt-HfO_2_	1000 °C	[17]
Sensor in this study	1300 °C/1100 °C > 60 min

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
