# Peer review of "Novel Multilayer SAW Temperature Sensor for Ultra-High Temperature Environments"

_micromachines, 2021, doi:10.3390/mi12060643_

Round 1
Reviewer 1 Report
In this paper the authors propose an AlN/Pt/Cr SAW temperature sensor realized in MEMS technology. They discussed the design and the manufacturing process of the sensors, showed experimental results by defining the sensors working temperature and sensitivity. Moreover an analytical model has been carried out by fitting the fitting the experimental sensor response.
The work is well organized and the obtained results consistent with the considered application. Despite this the reviewer has the following comments and concerns:
- The paper presents some typos and editing concerns such as, repetitions (line 222), missing space between value and units, please correct them.
- The caption for Figure 2 c) and d) are not exactly consistence with the figure, please describe it better.
- Line 200-220: wireless measurements are not presented or analyzed in the paper neither relating to the realized work, this discussion is forced and seems to be meaningless.
- In the reviewer opinion, based on the obtained results, it is not possible to assert the sensor “can measure temperatures up to 1300°C”; could be better to refer to 1300°C as a limit (break) temperature for the sensor since reliable results has been showed up to 1100°C.
Reviewer 2 Report
It’s a good work on high temperature sensing using langasite-based SAW resonator. Mostly, the presentation is honest. However, claiming the sensor to operate at 1300C does not make sense as the sensor was damages after the third test. Instead, authors could find a safe operating temperature at which the sensors can survive for longer. Also, the content related to antenna is completely irrelevant here. I understand the authors are trying to claim the potential of making wireless sensors but providing a paragraph about existing antennas does not support for that. The authors either want to come up with unique antenna types, their fabrication and testing at elevated/high temperatures or complete remove this part.
- Authors have presented the frequency vs temperature data and fitted the curves. I think Eq. 1 are the equations obtained from the fits in three different regimes. If yes, this need to be clarified. Also, did the authors calculate TCF1 and TCF2 from these curves? If so, how consistent are these values with the standard ones?
- Authors mention different temperature sensitivity at different temperature regime. For example, the resonant frequency first increases (up to 100C), then decreases, and the slope is larger for higher temperature. But, the physics/explanation is lacking.
- The purpose of using AlN films was mentioned once in the motivation but it was never demonstrated experimentally. Authors could have fabricated devices with and without AlN films and showed the value of the AlN films. Current write up is not clear about how AlN helped in improving the temperature stability. Also, is the AlN film non-piezoelectric here? This needs to be clarified.
- Fig 1(c) – LGS and AIN arrows point the same material.
- “However, the sensor cannot work many times at 1300°C.” (line 142, page 4) is not clear. What does it mean?
- How accurate the temperature is at low end? Usually, these furnaces (and sometimes the thermocouples) have variations when operating at large temperature window. I suspect the accuracy <300C in many large furnaces that are designed for high temperature applications.
- Evaluation of the sensor materials after testing by XRD is not sufficient for chemical composition. XPS is recommended.
Reviewer 3 Report
This is a well written article with solid results. It will certainly help the audience to understand the SAW sensor applications. There's no improvement or revision necessary for current edition.
Author Response
Thank you for your kind review comment
Round 2
Reviewer 2 Report
The revised version looks reasonable. Suggested changes have been made. I suggest to further refine followings:
Two specific points: (A) physics has been added to explain the temperature effect, however, the explanation is still lacking to address few questions such as why the resonance frequency increases in low temperature regime and decreases at higher temperatures.
(B) The number of times that can be measured at 1300C (or >1100C) is limited to 3. This basically tells the sensor is not stable beyond 1100C. In line 157 (page 5), authors could have written something like "Even though the sensor is able to measure up to 1300C, the sensor's survival time is limited beyond 1100C. The sensor becomes unstable (or dies) after three exposures to 1300C."
